# Behavioral Transition Path and Pivotal Nodes Regulating Attack in Initial Encounters between Unfamiliar Pigs

**DOI:** 10.3390/ani14172595

**Published:** 2024-09-06

**Authors:** Zhen Wang, Zhaowei Xiao, Hao Li, Zhengxiang Shi, Shihua Pu

**Affiliations:** 1College of Water Resources & Civil Engineering, China Agricultural University, Beijing 100083, China; wang@cau.edu.cn (Z.W.); sy20233092119@cau.edu.cn (Z.X.); leehcn@hotmail.com (H.L.); 2Chongqing Academy of Animal Sciences, Chongqing 402460, China; push@cqaa.cn; 3National Center of Technology Innovation for Pigs, Chongqing 402460, China

**Keywords:** attack, exploratory behavior path, dominance hierarchies, pivotal nodes

## Abstract

**Simple Summary:**

The intricate interplay of behavioral paths within animal groups, governed by dominance hierarchies, poses a challenge in identifying the pivotal node within the complex causal network of exploratory behaviors leading to aggression formation. Here, based on the strong curiosity of pigs and their competitive biological characteristics among conspecifics, two consecutive experiments were designed, aimed at investigating the behavioral transition paths and pivotal nodes governing aggression between two unfamiliar individuals upon initial encounter under controlled conditions. In Experiment 1, pigs that were mixed together showed short behavioral paths, primarily involving direct aggression, with a sequence of looking followed by attacking. In contrast, Experiment 2 involved adding new pigs to resident groups, leading to a more detailed sequence of behaviors. Resident pigs first engaged in a series of behaviors—looking, sniffing, touching—before transitioning to aggression. The study identified eight types of behavioral transition paths ranging from short, direct interactions to longer, more exploratory sequences. In both experiments, aggression was a common outcome, but Experiment 1 mainly showed brief exploratory pathways ending in aggression, while Experiment 2 revealed a variety of longer exploratory pathways involving multiple behaviors before aggression. This research highlights the complex nature of interactions between unfamiliar pigs and how their behaviors evolve during encounters.

**Abstract:**

This study leverages the inherent curiosity of pigs and their competitive nature among conspecifics to explore behavioral transition paths and critical nodes that govern aggression during initial encounters between unfamiliar individuals. Two consecutive experiments were designed to investigate these dynamics under controlled conditions. In Experiment 1, unfamiliar pigs engaged in one-on-one fights with quick retreats, displaying a simple behavioral sequence of looking followed by attacking. In Experiment 2, the addition of new pigs to resident groups resulted in a more complex and structured behavioral sequence. Resident pigs exhibited a ‘four-step’ exploratory behavior pattern: looking, sniffing, touching, and attacking. Further analysis revealed distinct exploratory pathways. In Experiment 1, only short behavioral paths were observed, while Experiment 2 revealed both long and short paths. Specifically, Experiment 2 uncovered seven types of behavioral transition paths, four of which were long and three short, highlighting different combinations of the basic behaviors. The transition paths involving aggression were more varied in Experiment 2 compared to Experiment 1. Overall, the 15 most frequent and obvious behavioral transition paths were identified across both experiments. Eight types of paths were categorized based on the transitions between the basic behaviors. These findings enhance our understanding of the behavioral dynamics in unfamiliar pig encounters, emphasizing the complexity of social interactions and the conditions under which aggression occurs.

## 1. Introduction

In natural populations, animal aggression is often considered a typical quantitative trait [1]. However, individual aggression is influenced by various social and physical environments, resulting in diverse behavioral manifestations [2]. Therefore, applying general conclusions from natural environments to scaled-up farming settings requires further refinement and adjustment.

Studies have indicated that aggression resulting from mixing is a behavior pattern animals adopt to re-establish dominance hierarchies, deemed inevitable in such contexts [3,4]. Consequently, most animal management guidelines recommend minimizing mixing events. However, in large-scale animal production systems, mixing is a practical necessity, such as in sow group-housing systems and in the mixing of piglets after weaning. If mixing is not possible, the welfare issues associated with sow confinement systems will be difficult to address [5,6]. And maintaining piglets in their original litters would require more building facilities and land, which is difficult to achieve in commercial farms. Therefore, it is necessary to conduct research to gain new insights into the characteristics of aggressive behavior during animal mixing.

It is generally believed that when resources do not meet the animals’ needs, conflicts within the group are inevitable. However, aggressive injuries can still occur even when resources are met, and the timing of such events cannot be accurately predicted in advance. Examples include big pig fighting, piglet tail biting, and cannibalism in chickens [7,8,9,10]. Currently, a commonly used control method in practical farming is ‘harm transfer’, such as tail docking and beak trimming [11,12]. The concern with this control method is that not using it results in unpredictable harm, while using it causes a one-time, definite, human-inflicted harm. Therefore, it is necessary to conduct research to gain a new understanding of the characteristics of aggressive behavior formed under these environmental conditions.

In natural environments, intraspecies aggression among group-living animals is limited and often follows ritualized procedures, rarely leading to life-threatening situations [13,14,15]. However, in managed environments such as farms, the stability of social groups can be compromised, and there is often limited opportunity for individuals to distance themselves from aggressors. This instability, coupled with high density and restricted space, can result in prolonged aggression, including biting, injuring, and even fatalities [16,17]. This highlights the need for a new explanation and effective control methods, as the behavior observed is influenced by the unique conditions of farm environments rather than solely indicating the lower social skills of the animals.

Is there a common characteristic in the formation of aggressive behaviors in farming environments or other possible contexts, such as intensive animal housing systems? This is the question that needs to be addressed.

Through systematic observation of numerous samples and cases, we have noted that aggressive behavior is neither a singular nor a basic action and may not follow a unified pattern. The formation of aggressive behavior does not result from a one-way causal relationship involving relatively independent components. We have realized that, at least in large-scale farming environments, aggressive behavior emerges from the transformation and escalation of various exploratory behaviors. Aggressive behavior is a member of the exploratory behavior system. The exploratory behavior system is more complex, fundamental, and widespread than the aggressive behavior itself, with the spontaneity of exploratory behavior leading to aggression. Therefore, we hypothesize that there is a central node in the causal network of complex exploratory behavior systems that contributes to the formation of aggressive behavior. Identifying this central node will help unify the understanding and control of aggressive behaviors that arise in different contexts. To this end, this study will focus on finding this central node along widely occurring exploratory pathways.

## 2. Materials and Methods

The study received approval from the Animal Welfare and Ethical Committee of China Agricultural University. All experimental procedures strictly adhered to the Guidelines on the Ethical Treatment and Welfare of Experimental Animals and followed the relevant regulations of animal welfare and experimental management at China Agricultural University.

### 2.1. Animal and Management

This study was conducted at a traditional fattening pig farm located in northeastern China (45°49′ N, 130°34′ E). This farm operates as an intensive closed-cycle facility, managing the entire life cycle of pigs from birth to market weight. The pigs used in this study were crossbred females of Landrace, Large White, and Duroc, along with castrated males. They were randomly housed together to ensure experimental diversity and representativeness. These pigs (n = 20_Experiment 1_ + 14_Experiment 2_), all from 18 (4_Experiment 1_ + 14_Experiment 2_) different pens of the same batch, weighed approximately 150 kg and were of a similar age.

During the experiment, a pig barn with adjustable pen sizes was selected within the farm as the venue for agonistic meetings among unfamiliar pigs. The dimensions of the barn were 12 m in length, 9 m in width, and 2.4 m in height, containing 8 adjustable-size pens. Each pen provided 3–5 square meters of activity space per fattening pig, adjusted according to individual pig space requirements. Environmental conditions were controlled to maintain optimal housing conditions, with temperatures set between 18 and 25 °C and relative humidity between 50 and 70%. Automatic drinkers were installed in each pen to ensure sufficient access to water, while pigs were provided ad libitum access to feed and water. Nutritionally, all pigs were fed a balanced diet according to the NRC (2012) recommendations, comprising yellow corn, wheat bran, soybean meal, corn oil, calcium carbonate, dicalcium phosphate, salt, L-lysine hydrochloride, L-threonine, L-tryptophan, DL-methionine, and vitamins. Additionally, roughage (straw and cabbage) was provided daily at 13:00 to enhance oral activity. To avoid any potential impact on pig attack, we did not use ractopamine or other growth-promoting additives. Throughout the experiment, pigs were monitored at least once daily to ensure free access to water and feed, as well as to monitor their overall health and well-being.

### 2.2. Experimental Design

To ensure the observation of the transition paths of basic behaviors when unfamiliar pigs meet within a limited time frame, we capitalized on the strong curiosity and competitive nature inherent in pigs [18]. Two consecutive experiments were designed for an in-depth exploration of behaviors and their dynamic evolution following initial encounters between any two unfamiliar individuals.

Experiment 1: At the beginning of the experiment, there were 8 pens in the barn, with 4 of them housing 4–6 pigs each, forming 4 separate groups. These groups were then paired by merging the pens, resulting in 2 larger groups, each containing 8–12 pigs. Following this, the 2 larger groups were gradually combined into a single group of 20 pigs, with all 8 pens merged into 1 large pen. This process occurred over 1–2 h to minimize stress and allow for acclimation. Experiment 2 commenced once the hierarchical relationships within the group were stabilized. Stability was determined by the absence of aggressive behavior within the group, which was observed by the seventh day post-mixing.

Experiment 2: The group from Experiment 1 was designated as the resident group. Each day between 12:00 and 13:00, a new pig was randomly added into this resident group, ensuring unfamiliarity among the new and resident pigs while maintaining comparable body weights. Familiarity at this stage was defined as sharing a previous pen with no regard to whether they were littermates. This process was repeated 14 times to ensure experimental reproducibility and to thoroughly observe and identify various behavioral transition pathways.

### 2.3. Framework for the Analysis of Animal Social Behavior

To better understand animal social structures and behavioral patterns, we have, for the first time, systematically summarized various behavior paths leading to aggressive behavior. We have also classified cases where dominance hierarchy information within the group is zero and non-zero, as illustrated in Figure 1.

In Experiment 1, we mixed unfamiliar pigs together, resulting in a situation where dominance hierarchy information was effectively zero. Since all pigs were strangers to each other, no pre-existing hierarchy existed. The interactions observed were primarily aggressive and confrontational, reflecting that the pigs were unfamiliar and had similar power levels at the start. This led us to classify the situation as one where the dominance hierarchy information was zero. The minimal power differences among unfamiliar pigs were inferred from the nature of their interactions and the lack of established social ranks. Additionally, this study highlights that the pigs in Experiment 1 were sourced from four different pens of the same batch, with all pigs weighing approximately 150 kg.

In Experiment 2, new members were added into an existing group (formed in Experiment 1, namely the resident group), resulting in non-zero dominance hierarchy information. When unfamiliar animals encounter each other, dominant individuals often exhibit curiosity, while subordinate ones experience fear and avoidance [19]. The dominant ones typically initiate aggression first. Resident pigs hold a dominant position compared to newly added pigs because they are familiar with the environment and have the advantage of numbers [20]. In contrast, the newly added pigs are usually fewer in number (in Experiment 2, only one pig was added each time). When these new pigs enter an unfamiliar environment and face a large group of unfamiliar pigs, they naturally exhibit fear and avoidance. The aggression was observed as group-level attacks on individual newcomers, highlighting significant power differences between the group members and the new arrivals. This phenomenon has been consistently observed in previous studies [20]. This classification reflects the presence of a stable dominance hierarchy within the established group.

We divided cases into two categories based on the presence or absence of dominance hierarchy information. Zero Dominance Hierarchy: this includes situations where individual differences in strength are negligible, as observed in Experiment 1 with unfamiliar pigs. Non-Zero Dominance Hierarchy: this includes situations with unstable or stable hierarchies, as seen in Experiment 2 with the integration of new members into an existing group. This detailed categorization provides insights into the dynamic relationships and mechanisms underlying exploration path formation within the group. By understanding these classifications, we can analyze the characteristics of behavior paths and aggressive behaviors in different social contexts.

### 2.4. Behavior and Its Pathway Analysis

Given the nature of this study, focal sampling was employed, allowing the observation of one pair of pigs at a time to ensure all relevant behaviors during each encounter were captured [21]. The sampling focused on specific behavioral events, such as sniffing, touching, and biting, which were critical to the study’s objectives. Although continuous filming was not feasible due to logistical constraints, this method effectively documented critical behaviors and transitions whenever they occurred.

To capture specific behaviors as they occurred, video recordings of each trial were selectively made using PHONE XR (APPLE), providing flexibility in documenting unpredictable and brief behaviors. Observers were trained to recognize and swiftly record these events, minimizing the risk of missing crucial interactions.

The recorded videos were subsequently analyzed using Adobe Premiere software (San Jose, CA, USA). The version used is Adobe Premiere 2018, and the official website is www.adobe.com. Each video was meticulously screened to identify and analyze agonistic interactions, termed encounters, between unfamiliar pigs. Encounters between unfamiliar pigs were defined as instances where the pigs were within one body length of each other and exhibited behaviors outlined in the ethogram. Table 1 documents ethograms of encounters between unfamiliar pigs in Experiment 1 and Experiment 2. To ensure consistency in defining the end of an encounter, a period of more than 2 s without any behavioral displays was used as a cutoff point. In each movie, we selected the pair of pigs with the most frequent and obvious behavior path. We coded a total of 15 pairs of animals in 15 movies. Each pair was unique, with no repetition among the observed pairs. This approach ensured that our observations were distinct and not influenced by repeated measurements of the same individuals.

Additionally, it is important to emphasize that the identification of paths in each encounter was based on capturing the most frequent and obvious path observed in each movie. To provide visual references, Appendix A are available on the journal’s website. In these movies, Appendix A captured the most frequent and obvious behaviors and their transition paths observed during encounters between unfamiliar pigs after mixing in Experiment 1, while Appendix A document the most frequent and obvious behaviors and transition paths exhibited by resident pigs toward newly added unfamiliar pigs when they first appear in the resident group in Experiment 2, specifically focusing on the initial encounters between resident pigs that notice the new arrivals.

## 3. Results

### 3.1. Behavioral Pattern of between Unfamiliar Pigs in Agonistic Meetings

By using Adobe Premiere software (Adobe Premiere 2018), video recordings of each trial were screened for agonistic meetings between unfamiliar pigs. From 15 movies of encounters between 2 unfamiliar individuals, we identified different types of behavioral paths in encounters observed in both Experiment 1 and Experiment 2. In Experiment 1, post-mixing, unfamiliar pigs engaged in one-on-one fights and swiftly retreated upon being overpowered. The behavioral pathway during this process primarily showed looking → attacking.

However, in Experiment 2, we observed that each time a pig was added to resident groups, the resident pigs that noticed the new member for the first time stopped their present behavioral responses to other stimuli and switched to a series of behavioral responses to the sudden appearance of the newly added pig (Figure 2). Firstly, resident pigs locked their eyes onto the newly added pig (see Figure 2a), and then sniffed the odor on the body of the newly added pig (see Figure 2b), then touched and prodded the newly added pig’s body with their noses (see Figure 2c). Subsequently, one resident pig was the first to open its mouth to bite the newly added pig tentatively (see Figure 2d), mostly on the head and shoulders. This process clearly showed a pattern of behavior from preliminary exploratory to in-depth exploratory, i.e., the ‘four–step’ exploration behavior pattern of looking → sniffing → touching → attacking.

When newly added pigs were bitten, some of them fought back (see Figure 3a). After failing to defeat the resident pig, they quickly sought refuge within the resident pig group to avoid further attacks (see Figure 3b) and subsequently urinated (see Figure 3c).

### 3.2. Behavioral Transition Paths between Unfamiliar Pigs in Agonistic Meetings

Subsequently, we analyzed the exploratory pathways formed by the basic behaviors of looking, sniffing, touching, and attacking. In Experiment 1, no long exploratory pathways including all four basic behaviors were observed; only short exploratory pathways were noted (Figure 4a). In contrast, Experiment 2 revealed multiple type of long exploratory pathways involving looking, sniffing, touching, and attacking in the behavior of the resident pigs towards newly added unfamiliar pigs (Figure 4b), as well as several type of short exploratory pathways (Figure 4c).

Specifically, in Experiment 1, one type of short exploratory pathway was observed: looking → aggression (Appendix A). In Experiment 2, seven types of paths were observed, comprising four types of long exploratory pathways and three types of short exploratory pathways. The long exploratory pathways included explicit transitions among looking, sniffing, touching, and/or attacking (Appendix A). The short exploratory pathways included explicit transitions among looking, sniffing/touching/attacking (Appendix A).

For long exploratory pathways, in Appendix A, we observed that the most frequent and obvious exploratory behavior transition pathway was looking → sniffing → touching → attacking. In Appendix A (between the 2nd and 3rd seconds), we observed that the most frequent and obvious exploratory behavior transition pathway was looking → touching → attacking. In Appendix A (between the 0th and 1st seconds), we observed that the most frequent and obvious exploratory behavior transition pathway was looking → sniffing → attacking.

For short exploratory pathways, in Appendix A (between the 18th and 35th seconds; between the 46th and 54th seconds), we observed that the most frequent and obvious exploratory behavior transition pathway was looking → attacking. In Appendix A, we observed that the most frequent and obvious exploratory behavior transition pathway was looking → sniffing. In Appendix A (between the 1st and 4th seconds), we observed that the most frequent and obvious exploratory behavior transition pathway was looking → touching. In Appendix A (between the 19th and 50th seconds), we observed that the most frequent and obvious exploratory behavior transition pathway was looking.

In total, 15 of the most frequent and obvious behavioral transition paths were observed across Experiment 1 and Experiment 2, comprising 8 types of paths formed by transitions between basic behaviors. Table 2 lists eight types of paths identified through Experiments 1 and 2. From Table 2, it is evident that Experiment 1 generated short exploratory pathways, whereas Experiment 2 produced long/short exploratory pathways. However, both Experiment 1 and Experiment 2 observed aggression within the formed paths.

### 3.3. Distribution of Exploratory Pathways and Behavioral Outcomes in Pigs

Table 2 shows that Path 1 has one instance, Path 2 has three instances, Path 3 has one instance, and Path 4 has four instances. Among these four types of paths, totaling nine instances, no biting behavior was observed. Instead, these paths concluded with ritualized outcomes. Specifically, in these cases, pigs exhibited non-aggressive interactions. These ritualized behaviors are typical in situations where pigs establish a social order and relationships without resorting to direct aggression. Path 5 has three instances, Path 6 has one instance, Path 7 has one instance, and Path 8 has one instance. Among these four types of paths, totaling six instances, biting behavior was observed, and they ended with aggressive outcomes. Among the four types associated with attack, totaling six instances, one type originates from Experiment 1 and three types originate from Experiment 2.

## 4. Discussion

Identifying a pivotal node for the formation of aggressive behavior from the intricate causal network of exploratory behavior is a monumental challenge, yet pivotal for a unified understanding and control of aggression. Experiments 1 and 2 capitalized on the highly curious nature and competitive dynamics among pigs to anticipate diverse behavioral paths within a limited timeframe.

Experiment 1 simulated the process of group mixing, specifically focusing on the interaction between two groups of unfamiliar pigs being combined to form a larger group. Results showed proactive engagement in mutual exploration activities by individuals from both groups, identifying a short exploratory pathway from looking → attacking. Experiment 2 further explored the exploratory pathways and aggression nodes under conditions of pronounced dominance differences. The results showed that individuals from the resident herd typically exhibited dominance, while newly added pigs often displayed subordinate behavior. Dominant individuals engaged in longer exploratory pathways characterized by a long sequence from looking to sniffing, touching, and finally attacking. These findings mean that the comprehensive exploration process exhibited a hierarchical escalation: basic exploration → overlaid exploration → deep probing. Basic exploration involved a visual fixation on conspecific targets until the end of exploration; overlaid exploration added olfactory and tactile behaviors onto the fixation level; deep probing involved biting behaviors as forms of aggressive damage.

In this study, we identified 8 types, a total of 15 of the most frequent and obvious exploration pathways of varying lengths formed by transitions between 4 basic behaviors: looking, sniffing, touching, and attacking. Among these types of paths, four culminate in ritualized endings while the other four end with attack behavior. These ritualized behaviors exhibit non-aggressive interactions, which facilitate social bonding and hierarchy establishment. Such findings align with the existing literature indicating that pigs often use ritualized behaviors to establish social order and minimize direct aggression [23]. Conversely, the remaining four types of paths end with attack behavior. These paths involve direct attack, such as biting or fighting, which serves as a mechanism for establishing dominance and resolving social conflicts. The presence of these aggressive endpoints highlights the role of competitive interactions in the social dynamics of pigs, particularly in situations where ritualized behaviors are insufficient for resolving conflicts [4,13,24].

Within animal communities, conflicts arise when resources such as food, water, and space are temporarily insufficient or cannot be simultaneously provided, indicating that the order of utilizing limited resources within the group has not been fully established, and hierarchical relationships remain unstable [25]. Evidently, short exploratory paths awaiting escalation still exist in such scenarios. Under conditions of abundant resource supply, mutual exploration remains at basic and cumulative levels, without the need for further escalation. However, in cases of resource scarcity, animals unable to access food promptly must weigh the consequences of hunger against the potential harm of challenging the established order. Since attacks during feeding are typically one-on-one interactions with minimal harm inflicted even in the event of a failed challenge [26,27,28], individuals in urgent situations may skip the intermediate steps and directly resort to aggression to make a final determination. This explains why a short behavioral transition path from looking → attacking was observed in Experiment 1.

Even in situations where individuals are familiar with each other, dominance hierarchies are stable, and food and water supply are abundant, frequent aggressive behaviors still occur within animal groups, such as tail biting in piglets [9], ear (tail) biting in larger pigs [7,10] and vent pecking in chickens [8]. These aggressive behaviors are challenging to observe through experimental design due to their unpredictable occurrence. However, when the environment lacks necessary external stimuli, individuals with high curiosity show a strong interest in specific body parts of conspecifics, leading to aggression [29,30,31,32]. It is plausible that exploration driven by curiosity is initially conducted in a relatively relaxed manner, and if aggression arises, it could follow an exploratory path that might involve looking → sniffing → touching → attacking. This proposed pathway should be considered as one possible scenario, recognizing that aggression is a complex phenomenon with multiple contributing factors [33,34].

Attention marks the beginning of the exploration phase, while aggression signifies its end. In our study, attention, sniffing, touching, and aggression are all considered forms of exploratory behavior. Sniffing, touching, and aggression can occur independently or simultaneously with attention. Attention serves as a prerequisite for other exploratory behaviors, which are influenced by the social dynamics of the pig group, such as hierarchy, dominant subgroups, or the focus on individual pigs versus specific body parts. Aggression itself constitutes a form of exploration, and once it concludes, exploration ends, as no further exploratory behaviors towards the object have been observed in production practice. Our previous research supports this view, showing that aggression provides a thorough understanding of the object, eliminating the need for additional exploration [20]. It is important to clarify that aggression is not merely a subsequent behavior but rather a significant form of exploration. Similarly, while touching can precede aggression, both behaviors can occur independently or together. Aggression, when observed, typically concludes the exploration phase, as we have not documented exploratory touching occurring after aggression. This understanding aligns with our observations that once aggression is complete, the exploration phase is effectively concluded.

## 5. Conclusions

In this study, we identified distinct behavioral patterns and transition paths between unfamiliar pigs during agonistic encounters. Experiment 1 highlighted a short exploratory pathway of looking → attacking in scenarios where unfamiliar pigs engaged in direct confrontations. In contrast, Experiment 2, which involved introducing new pigs into established groups, revealed more complex and prolonged exploratory pathways, often involving looking, sniffing, touching, and attacking. The analysis of behavioral transition paths revealed eight distinct types, with some paths culminating in ritualized, non-aggressive interactions, while others led to aggressive outcomes. The findings suggest that while pigs often use non-aggressive behaviors to establish social order, aggressive interactions are more likely when ritualized behaviors are insufficient for resolving conflicts. Overall, this study underscores the importance of understanding the intricacies of behavioral transitions in pigs, as they play a crucial role in shaping social dynamics within groups. The identification of specific exploratory pathways provides insights into how pigs navigate social interactions, balancing exploration and aggression based on the context of their encounters.

## Figures and Tables

**Figure 1 animals-14-02595-f001:**
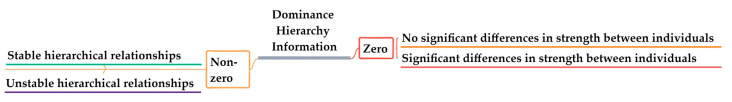
Methods for classifying dominance hierarchy information. To enhance our understanding of animal social structures and behavioral patterns, various paths leading to aggressive behavior have been systematically summarized for the first time. Furthermore, dominance hierarchy information within the group has been meticulously classified.

**Figure 2 animals-14-02595-f002:**
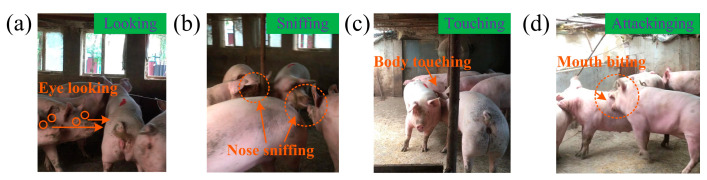
Ethogram of resident pigs encountering the newly added pig. (**a**) Resident pigs locked their eyes onto the added pig; (**b**) resident pig sniffed the odor on the body of the added pig; (**c**) resident pigs touched and prodded the added pig’s body with their noses; (**d**) one resident pig opened its mouth to bite the added pig. For high resolution, see Appendix A.

**Figure 3 animals-14-02595-f003:**
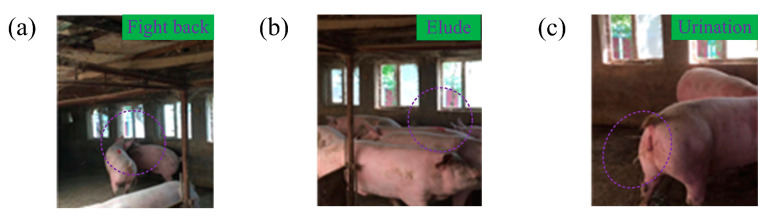
Ethogram of newly added pigs responding to resident pigs. (**a**) Newly added pigs fought back; (**b**) newly added pigs hid within the resident group; and (**c**) newly added pigs urinated.

**Figure 4 animals-14-02595-f004:**
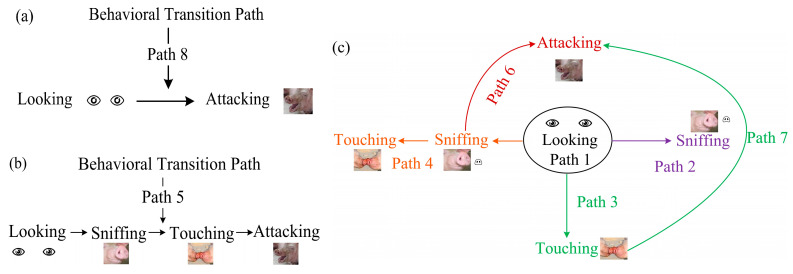
Behavioral transition paths between unfamiliar pigs in agonistic meetings. (**a**) The short exploratory path observed in Experiment 1; (**b**) the longest exploratory path observed in experiment 2; and (**c**) several type of short/long exploratory paths in Experiment 2.

**Table 1 animals-14-02595-t001:** Ethograms of encounters between unfamiliar pigs in Experiment 1 and Experiment 2.

Designation	Behavioral Pattern
Initial Encounter	The first instance where two animals begin to exhibit behaviors indicating mutual awareness or curiosity about each other.
Looking	The act of pigs maintaining visual focus on a target of interest for a sustained period, typically defined as lasting more than 3 s without significant distraction or interruption [22].
Sniffing	Behavior where animals use their noses to discern the scent of objects.
Touching	Animals’ exploratory touching of targets with their noses.
Attacking	Behavior where animals use their teeth to attack a target.
Fight Back	Animals’ response to an attack from another by retaliating.
Elude	Behavior where animals steer clear of aggressive entities.
Panic	Behavior exhibited by animals due to fear of attackers, such as urination, trembling, and rapid breathing, among other abnormal behaviors.

**Table 2 animals-14-02595-t002:** Distribution and characteristics of exploratory pathways, including aggressive and non-aggressive outcomes, in pigs. Note that Experiment 1 simulated a scenario where unfamiliar pigs encounter each other, while Experiment 2 simulated a different scenario of unfamiliar pig encounters, resulting in a total of 15 of the most frequent and obvious behavioral transition paths being captured in 15 videos. The numbers following each behavioral transition path do not represent the frequency of occurrence of that path, but rather indicate in how many of the 15 videos that path was observed. Additionally, it is important to emphasize that the identification of these 8 types of paths was based on capturing the most frequent and obvious path observed in each video.

Experiment	Path	Behavioral Transition Path	Number of Videos
2	1	Looking	1 (including Appendix A)
	2	Looking → Sniffing	3 (including Appendix A)
	3	Looking → Touching	1 (including Appendix A)
	4	Looking → Sniffing → Touching	4 (including Appendix A)
	5	Looking → Sniffing → Touching → Attacking	3 (including Appendix A)
	6	Looking → Sniffing → Attacking	1 (including Appendix A)
	7	Looking → Touching → Attacking	1 (including Appendix A)
1	8	Looking → Attacking	1 (including Appendix A)
Total			15 (including Appendix A)

## Data Availability

Data are contained within the article or Appendix A: The original contributions presented in the study are included in the article/Appendix A, further inquiries can be directed to the corresponding author.

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
