# Peer review of "Behavioral Transition Path and Pivotal Nodes Regulating Attack in Initial Encounters between Unfamiliar Pigs"

_animals, 2024, doi:10.3390/ani14172595_

Round 1

Reviewer 1 Report

Comments and Suggestions for Authors

General Comments –

This manuscript by Wang et al. addresses questions related to the establishment of social hierarchy in pigs as it related to agonist behaviors between conspecifics.  The paper could be improved by addressing the following concerns. 

1 – It appears to be a purely descriptive study without a clear hypothesis tested.

2 – No statistical analysis is presented for the reader to evaluate the significance of the observations (eg different from chance)

3 – Work appears to be based on the observation of a total of 22 behavioral transitions if I understand the paper correctly.  Most behavioral papers would have 100 or 1000’s of behavioral observations to address the inherit variability in animal behavior.

4 – No consideration is given to the impact of individual differences in animal behavior associated with different coping styles or personality types.

5 – Several aspects of the methodology is described only qualitatively and thus would preclude replication of this work by others.

6 – Accepted/Standard description of video coding is absent

7 – Several opportunities where reference/influence of previous work should be better incorporated into this manuscript including ethograms and Markov Models.  

Detailed comments are provided below to help the authors address these concerns.

Specific Comments

L25 – Why is Simple Summary identical to Abstract?

L49 – Relevance of insect behavior references to pigs need to be better developed or explained.

L55 – “minimizing grouping” – do you mean mixing events or size of groups?

L57 - Makes no sense, individual housing prevents sow aggression.

L59 – “original pens”  - Same pens or litter groups.  Same pens really unlikely solution of commercial farms

L60 – Relevance of human communities to this discussion is unclear.  Also should be cited if there is a justification

L62 – How do “rank sequences” meet a need of an animal?

L66 – Controlled vs uncontrolled injury?  What is a controlled injury sounds like willful neglect.  Believe there are better words to describe this concept.

L99 – Were pigs feed ad lib as elsewhere in the paper suggests otherwise.

L115 – What is meant by gradual?  Not a specific or precision term.  What was the time frame of this mixing event.  Methodological details need to be explained such that it would be possible for other to repeat the experiment.  The manuscript falls short here on this account as well as elsewhere in the paper.

L121 – What objective criteria was used to know that hierarchy was established in experiment one

L123 – What was the time between the end of experiment 1 and the start of experiment 2

L124 – “1-2 hours after the 2nd feeding” seems to contradict ad lib feeding statement in methods

L141 -  how “was dominance hierarchy information within the group zero” “meticulously classified”?.  Again what objective criteria were measured and used to make this distinction?

L148 – how as the “the difference in strength between individuals” measured objectively?  Sorry these kinds of statements are not acceptable in the methods section of a scientific paper reporting empirical results. 

L166 – Methodological details of video coding is woefully insufficient here. Authors are suggest to refer to Bateson and Martin – Measuring Behavior for standards in study of ethology.    

L181 – Note the application of Markov models has been applied previously to describe behavioral transitions in animals (Rugg & Buech, 1990; Macdonald & Raubenheimer, 1995) and most recently to pigs (Ede et al, 2024).  In most cases, these applications have used  Markov model outcomes to drive Monte Carlo simulations that allow for testing the statistical significance of specific transitions.   This might be a helpful approach here as we are left with the question of what is the significance of the data you report.  Regardless, the previous work should be cited and any novel applications in this paper explained.  

L212 – “trends of pig group behavior” – Recognize that any “group” behavior is really the aggregate of many individual animals making assessments and behavioral responses.  The notion of a group behavior is likely most appropriate if the animals under study come from a normally distributed population in terms of behavior.  However, more and more data is emerging on pigs that show individual differences in behaviors and personality types and that these difference can be orthogonal i.e. proactive vs reactive coping styles etc.   Many groups are incorporating simple personality tests like open field and or novel object testing into their experimental designs to help test these underlying assumptions about the study animals. 

L213 – Is this really statistical analysis or model analysis?  As I read the work it is descriptive in nature and the reader is left wonder what assurances do we have if this work was repeated over and over again how often would these findings be observed.   If there is hypothesis testing to some level of statistical significance for meaningful interpretation here then it needs to be explained in more detail.

L234 – What does “extensive” mean? It is a qualitative statement to which the authors can and should provide a quantitative answer – eg how many minutes or hours of video was coded in this study?

L235 – “between unfamiliar pigs” – clarification in the methods is needed regarding what animal(s) is being studied.  Is it only the behavior of the resident animal that is being observed and if so then what was coded in Exp 1 before a resident group of animals was identified. 

L236 – Is it ok to include behavioral transitions in the same analysis if the underlying experimental conditions are significantly different to preclude expression of certain behaviors?  Isn’t an underlying assumption in MM is that the probability of transitions are stable/stationary?

L236 – “Ethogram” – Usually ethograms are defined in advance of the study to ensure that the behavioral observations are mutually exclusive and objective.   Potentially confusing use of term here to describe an outcome of the study.

L239 – Why is the behavior in exp 1 so different than exp 2 with the lack of  olfaction and touching.  Is it methodological as it is unclear how animals were recorded and coded.  Are the behaviors just faster and missed?  As an aside olfaction may not be the best term here as the pigs can smell for a distance, what you are observing is more sniffing or nose touching. 

L247 – Does the resident pig always bite first and how would be know if you are only coding the behavior of the resident animal?

L252 – Nice to include photos for ethogram.  However, they are small and sometimes day.  Consider including in larger format in supplemental information.

L262 – Ethograms are usual presented in the methods.  There are several existing ethograms in the literature for aggression between pigs.   Would be good to tie your work to the existing body of knowledge and explain why an new or different definitions of behavior are needed in this paper. 

L265 – What is the justification for terms like discontinuous or incomplete?  Maybe not all these behaviors are required for a pig to achieve their intended behavioral goal.

L280 – How many total animals (or total pairs of animals depending upon how you defined your coding strategy – which was not provided) were observed in this study and were they unique?

L287 – Again why is it incomplete if it ends in an attack?

L291 – Table 2 – Experimental number labels don’t make sense and seem to contradict other parts of the text.  Experiments 2 = Path 8 = Attention to Attack = 1, but I thought Attention to Attack was primary behavior sequence in Exp 1?

Related to that, isn’t attention to attack the most common event in exp 1, then why is there only 1 observation of it in the whole analysis?

L291 – Just to be clear all the analysis on attack is based on 6 transitions as reported in Table 2 (sum of amount for Path 5 to 8)?

Furthermore over half the pathways (1,3,6,7,8) were only observed once? What is the significance of these pathways if they only occurred once in the study?

L294 – What is the 9th exploratory pathway as only 8 are described in table 2?

L313 – As mentioned above do you worry about individual differences in behavior between animals and is that accounted for in your analysis.

L315 – What is “prolonged” – Again unacceptable qualitative term.  How long were the simulations?

L316 – Just unclear to me if describing the steady state distribution of this model makes any sense in the context of the problem being studied.   The data is based on observation of single, initial encounters of pig pairs following a mixing event.  Running the model to equilibrium would be to iterate these probabilities of transitions without the resetting event of introducing new animals.  Seems that MM is a useful tool for capturing the initial dynamic of behaving animals but to run it to equilibrium is less intuitive to me, but maybe more explanation can clarify why this is important. 

L329 – Again unclear what information was coded from the videos.  However, if the authors have the duration times of each behavior for each animal that would be interesting to report or if not then the authors need to explain more where the average residence time comes from and what it is and what are its units.  

L330 – What do you mean by “extended time scale” again aren’t these behavioral sequences trigger by a mixing event.  Once the hierarchy is established won’t the transition probabilities change.  Hard to understand how model can predict this.

L333 – Here the number of attacks is 8 but the number of paths with attacks is 6.  Is this an inconsistency in the data reporting or are their pathways with more than one attack event and then how does that happen as attack seems to be a dead end event in the behavioral sequence.

L334 – Just to be clear this whole paper and analysis is based on the observation of 22 behavioral transitions and there is no statistical significance of these observations reported?

Comments on the Quality of English Language

Word choice is somethings unusual.  Could be improved in reviewed by native English speaker with some understanding of the field.  

Reviewer 2 Report

Comments and Suggestions for Authors

This study addresses a significant issue in animal husbandry: the aggression in pigs during integration into new groups. This is a critical area of research due to its implications for animal welfare and farm operations. The study is well-designed, the data analysis is thorough, and the findings have notable implications for animal welfare in pig farming. Additionally, this study identifies specific factors and methods that should be considered when mixing pigs in production practice. Overall, the manuscript provides valuable insights into managing aggression in pig herds and enhancing animal welfare. Researchers shall quote more domestic and foreign literatures related to conflict

Comments on the Quality of English Language

Minor editing of English language required

Reviewer 3 Report

Comments and Suggestions for Authors

General comment

The study is very interesting in theory and used appropriate methods of analysis. I truly appreciate that the authors shared their videos and codes, which helped a lot understanding the study and the analysis. However, I have some big concerns about the data collection. At the moment, the value of the paper is quite low, but it could be improved with a better description of the methods and the results, of the terms used.

The methods need to be more details, at the moment there are a lot of important information missing about how the videos were recorded (material and duration of recording), the assessment of the hierarchy and of the way experiments were conducted.

From the supplementary material, it is visible that the videos (as it can be seen in the supplementary material) were recorded using a camera held by an observer who was standing and moving in the pen. The target animal was followed by it is evident that there is a loss of information due to the loss of focus (e.g. when the pig or the human moves). Therefore, I believe that such video material is not suitable for the type of analysis that the authors wanted to perform (i.e. Markov chain analysis necessitates that the behaviours are continuously observed). At minimum, this methodological flaw needs to be addressed in the discussion - along with a recommendation to further studies to use continuous recordings from fixed cameras.

Deep probing really needs to be described, as it is used throughout the manuscript but does not relate to a term commonly used.

I am also concerned about the results: it is reported that, out of two experiments and over 15 mixing of pigs, only 15 events (Table 2) were observed (or 22, as reported in Tables 3/4), which is VERY low, and insufficient to run Markov analysis. If more events were observed, the total number of occurrences needs to be presented. 

Finally, I have doubts about the relevance of the references used. A lot of them are about humans, or species very unrelated to pigs (e.g. drosophila). I would encourage the authors to focus more on the pigs and other farm species to support their statements. General comments about the in-text citations: it is better to place the reference next to the concerned species/statement, in the case of citing multiple studies.  

See specific comments below.

Introduction

L57-58: I am not sure about that statement. In general, more aggression is reported in group-housed sows, compared to sows housed in single stalls. Group housing is better for the welfare of the animals as it allows a social life, but it also comes with more aggression, especially when groups are made again (e.g. after weaning).

L59-60: I think the comparison with human societies is not relevant here. Delete.

L65: What does "controlled" and "uncontrolled" injuries refer to? Please define them as those terms are not usually used.

L69: The problem of prolonged aggression is rather that in farm environment, the groups are not stable, there is no opportunity to distance / run away from the aggressor. Saying that "dominant individual do not cease aggression" suggest that the animals have lower social skills in farming environment, which is not necessarily true.

L72: which circumstances do you refer to? Farming environment? What are the other scenarios?

Material and Methods

L98: do you mean "drinkers" instead of "waterers"?

L115-118: That does not make sense. You had four pens, then merged two (so you have four) and then incorporated the remaining two (so you have one).

Do you mean that you first paired pens (so making two bigger groups out of the four initial groups), and then merged these two bigger groups into one (of 20 pigs)?

L118: Maybe state the characteristics of the pigs (age, weight, sex and breed) before the description of the experiments.

L120: detail how hierarchy stability was assessed, and how long it took to reach it.

L128: was the unfamiliar pig then taken away from the group or did it stay with the group?

Please specify.

if 34 pigs were kept in half of the barn, then pigs in this group had only 1.5 m2 space

L130: what do you mean by "free-range”? Did pigs have access to an outdoor space? If so, please detail the size of the outdoor space.

L133-135: I believe that the ethical statement should be at the beginning of the Material and Methods section.

L137: You have to describe how the behaviours were recorded: what type of camera and angle, for how long, how were residents and unfamiliar pigs identified on camera?

L141-142: This requires a lot more details to be understandable. What does "dominance hierarchy information" refer to? How did you assess/measured it? From the figure, I understand that "zero" is about the strength differences between the individuals, while "non-zero" is related to the stability of the hierarchy relationships. It is not clear why you used a binary score for "dominance hierarchy information" to describe two different aspects that characterises it. I would expect to have a score (binary) for each aspect (i.e. strength difference and stability of relationships), and to have a global score (0-2) for the "dominance hierarchy information".

L143-159: You need to describe how you assessed the difference in strength and the stability of the hierarchy. At the moment it seems arbitrary and cannot be reproduced in further studies.

Results

L234: footage, not foot-age

L238: retreated, not re-treated

L252: I think that emotions should not be interpreted here. It could be fear, stress, or also activation of the elimination system due to intense activity, or release from holding elimination need due to previous fight.

Figure 3: I think picture b and picture c are confounded: picture should be "c" as it shows a pig urinating, and picture c should be b as it shows a pig hiding within the group of pigs.

Table 1:

How do you define "interest in each other”?

what did you consider as "continuously observing”? Was there a duration limit?

L271-279: The reference to the supplementary videos is very nice. However, it would be better to either state the time of the video at which the sequence occurs, or to indicate it in the video (e.g. with an arrow, circle, or simply adding text to the video to narrate the sequence).

Table 2:

1. in the table, "Experiment 1" and "Experiment 2" (singular, not plural).

2. Experiment 1 and experiment 2 have been inverted: according to Figure 4 and the text, experiment 1 has path 8 only, while experiment 2 has paths 1-7

L294-299: please revise the text and the exactitude of the wording: you mention that there are "nine paths" (L294), then "six paths" (L295). Then you refer to the "six paths including attack" (L296 and L299) but Table 2 only presents four paths (and you mention four paths at L295).

This also caught my attention: what do you call an "exploration path”? Because Path 1 and Path 8 only include "Attention" (and "Attack" for Path 8). And what do you call "ritualized endings”? Please provide a description on the endings when attack is not involved, that would support naming them "ritualized".

L300: How can you conclude that the attack in Experiment 1 occurred near the conclusion of the exploration phase? Path 8 only mentions "Attention" followed by "Attack", so this is the beginning, and not the end, of the exploration phase.

L303: again, the statement that Attack occurs at the end of the exploration phase does not make sense, out of four identified paths concluding in an attack, only two Paths (5 and 7) included "Touch", the other two do not involve a physical exploration of the unfamiliar pig.

L306: "deep probing" is not supported by any of your results, only two paths involved "touch" before "attack", and "deep probing" carries the meaning that the exploring pig was insistent, maybe forceful towards the unfamiliar (explored) pig.

L315: The sentence needs to be reworded - "Results from prolonged simulations revealed..."

Tables 3 and 4: I would argue that the two tables display the same information. Maybe only use Table 4.

L340: you conducted two consecutive experiments, or a series of two experiments, but not two consecutive series of experiments.

L342 and L344: "dominance hierarchy is (non-)zero" does not mean anything outside the context of your study and is not described in a comprehensible manner in section 2.3. Therefore, please describe it in section 2.3 AND in the discussion (the reader should not have to report again to the methods to understand what you mean in the discussion)

L347: same as above for "ritualized endings", you must describe what it means.

L349-354: This is not a result from your study. You studied the behaviour of the resident pig towards the unfamiliar pig, not the contrary. Otherwise, this is not clear at all in your results section, and you need to present those results to be able to discuss them.

L358-360: This is far-fetched... what experiment 1 simulated is group mixing, whereby two groups of pigs are mixed together to form a bigger group (e.g. what happens at weaning when litters are mixed, or when pigs change facilities in specialized production systems).

L363-365: This is absolutely not clear in your methods. You mentioned that the unfamiliar pigs were matched for weight, and there was no assessment of their hierarchy ranks. Thus, I do not understand how "more pronounced dominance differences" were obtained, and I would doubt of the accuracy of the statement as you added one pig to a group of 20 pigs (thus, the chance that the unfamiliar pig was very different to all other pigs is very low...). What experiment 2 did was to simulate the modification of the group structure in a less dramatic way than group mixing (e.g. what happens when weights are equalized and some pigs are exchanged between pens, or when a pig is reintroduced to the group after being in a "hospital pen").

L366: There is no mention of measuring dominance exhibition in the material and methods, or in the results. What are the subordinate behaviours recorded?

I see the dominance is inferred from exploration, but I would argue that this could be also a different motivation (i.e. sociality, exploration) than the affirmation of dominance.

L394-395: place citations next to the relevant species.

L399: I think saying "clearly" is too affirmative for the speculation, aggression is a very complex phenomenon.

conducted, not con-ducted

L404+L406: the affirmations that aggression arises from deep probing are wrong. in half of the exploration path that you identified; the attack occurred after a non-tactile event... thus no "deep probing" was involved before the attack.

L410-413: This is a very far-fetched statement... I would recommend deleting it as it does not add anything meaningful to your discussion, and it is not supported by scientific investigations.

Comments on the Quality of English Language

See specific comments

Reviewer 4 Report

Comments and Suggestions for Authors

In general, this is a good and thoughtful review of the sequence of events that can occur in interactions between pigs that may or may not lead to aggression., However,  suggest that it suffers from over-presentation of the analysis that becomes incoherent, especially section 3.4 on Markov analysis, (lines 307-332).

Table 2, a simple summary of the observations is clear. Table 3 presents the same data under the heading 'Frequency of transitions' This table, as defined is meaningless, since it does not describe transitions, it simply presents the numbers from Table 1 that proceeded to each stage and no further. The number 1 under Attention is completely meaningless.

Table 3 can become the new Table 2 since it expresses the same data as probabilities. However all the zero values that begin each line are not expressions of probabilities so should be deleted.

l 146 and Fig 1 'When dominance hierarchy between individuals is zero' we have divided classes according  to whether 'differences in strength' are significant or not significant' I can find no definition of strength or of how you measured the significance of differences.

Round 2

Reviewer 3 Report

Comments and Suggestions for Authors

I thank the authors for addressing all my comments, and the quality of their revision and replies. I believe that the manuscript is much improved and is now suitable for publication. 

I would just like to alert on the fact that the track changes in the document seem to have gone wrong, and that parts of the old version and the new version co-exist as some places... I advise that you and the editor check this together.